# Typical Fragment Kinetic Energy Assessment Based on Acoustic Emission Technology

**DOI:** 10.3390/s22155914

**Published:** 2022-08-08

**Authors:** Fei Shang, Liangquan Wang

**Affiliations:** School of Mechanical Engineering, Nanjing University of Science and Technology, Nanjing 210094, China

**Keywords:** evasion, acoustic transmit signal, wavelet analysis, kinetic energy assessment

## Abstract

Fragment kinetic energy is an important parameter to characterize the damage power of fragments. In this study, an acoustic emission technology-based method to evaluate fragment kinetic energy is proposed. The dynamic response of the fragment impacting an aluminum alloy target plate and the relationship between the initial kinetic energy of the fragment impact and the acoustic emission waveform were theoretically evaluated; the numerical simulation of typical spherical fragments (8 mm diameter) penetrating the aluminum alloy target plate was performed, the wavelet energy of the acoustic emission signal was obtained using wavelet packet theory, and a mathematical model of wavelet energy and fragment kinetic energy was constructed. A fragment kinetic energy test system was established, and a fragment penetration test was performed. The analysis showed that the wavelet energy mathematical models and the fragment kinetic energy exhibited favorable consistency, and the measurement errors of the three experiments were 3%, 3.7%, and 3%. This demonstrates the effectiveness of the typical acoustic emission fragment kinetic energy test methods proposed in this study and establishes a new method for the direct measurement of fragment kinetic energy.

## 1. Introduction

The role of breaking the film is significant in the battlefield. The process of destroying the damage is to obtain high-quality kinetic energy instantly under the influence of the blasting of the film, during emission at high speed. The impact and thorough target complete the killing of the target. The damage of the broken film is that its essence is the damage caused by the kinetic energy [1,2]. Therefore, a measurement method for a new explosion field is of military significance for damage evaluation and actual applications of the destroyer of the battle.

Currently, research on cutting kinetic energy is focused on the measurement of film-fragment speed. The initial speed of the film fragment is measured to calculate the clip speed attenuation coefficient, which is subsequently used to calculate the approximate value of the broken film speed. Typical methods to measure the tablet speed are of two types: contact and non-contact measurements. Contact measurements include aluminum foil targets, combed targets, and net targets. Non-contact measurements include light curtain targets, sky curtain targets, radar speed measurement methods, and high-speed photography [3,4,5,6]. When the film fragments are emitted, they are accompanied by strong impact vibration, high temperature, high pressure, and electromagnetic radiation. The aforementioned method is time-intensive. Moreover, this method is used to measure the average speed of the flight trajectory of the film fragment [7]. Therefore, it is required to theoretically develop a new testing method to break the film-breaking energy and the speed-killing effect of the broken film.

Recently, with the expanding applications of sound launch technology, the process of stress wave changes was induced in the structure during impact, turbulence, leakage, and injection [8,9,10]. Acoustic emission (AE) is defined as one or more local sources in the material to rapidly release energy and emit transient elastic waves [11]. It is essentially a non-permanent deformation of the transmission of high-frequency force waves in the medium. Rubic contact, impact, and thermal deformation that can cause structural changes, and the formation and growth of cracks in the same sex and orthogonal heterosexual materials, can be used as AE sources [12]. Wolfinger used a hammer to impact the composite material board, and the average root of the definition signal was considered as the valid value; moreover, the experimental curve of the effective values, impact energy, and impact speed were analyzed, and the possibility of impact quantitative assessment was mentioned [13]. Greszczuk et al. proposed the use of the Hertz contact theory during low-speed impact to derive the relationship between the impact and structural internal stress wave during impact [14]. Yang et al. used the wave lability of small-wave technology and energy entropy theory to analyze sound-transmitting signals during the impact of a hard object on an air compressor blade, and the relationship between the small wave bag energy entropy of the wavelet of the impact speed and sound-transmission signal was obtained [15], providing a new method for impact-monitoring engine blades of sound-launch technology. Hesser et al. used steel balls of different diameters to stimulate aluminum plates with voltage sensors for impact sources, which are based on convolutional neural networks and in-depth transfer learning methods; thus, the degree of damage to the AE source was quantitatively assessed [16]. Barile et al. used the sentry function to correlate response and sound energy to determine the damage dissemination and material degradation information of the composite materials [17]. Prosser et al. studied the transmission characteristics of transmitting waves under ultra-high-speed impact and observed that the frequency peaks of ultra-high-speed impact sound transmission signals change with impact speed. Low- and ultra-high-speed impacts both cause high board waves [18]. Runqiang analyzed the energy characteristics of the fragmented clouds of ultra-high-speed impact aluminum alloy plates in a low speed range and studied the dissemination of the projectile and the interaction of the pills and thin target board during impact; moreover, the quantitative relationship between energy dissipation and impact conditions during the ultra-high-speed impact was established to calculate and evaluate the fragmented cloud ginseng number [19]. Gu, through numerical simulation, observed that the A0 modal value in the high-speed impact sound transmission signal was proportional to the projectile impact. Based on this, the quality of the impact parameter of the projectile was estimated through the signal feature threshold of the sound transmission signal [20].

To summarize, the impact of the body, the momentum, and the effective value, of the sound-transmission signal, energy entropy, amplitude, and other parameters, were limited to the impact damage of various materials. The sound launch technology has not yet been used to conduct in-depth research on the quantitative calculation of cracking kinetic energy. To solve the problem of using sound-transmission technology to measure kinetic energy, this study numerically simulated typical fractures on high-speed impact aluminum alloy target boards and recorded the sound-transmission signal generated by the impact. Parameter analysis and wave bag theory analysis were performed to determine the relationship between dynamic kinetic energy and sound-launch features, and the target board test was verified using the measured sheet.

## 2. High-Speed Impact Dynamic Response of Thin Targeting

In the target board system of the broken film, the board and film fragments were deformed. This study proposed a simple energy balance model. The energy balance model is the energy equation generated using the dynamic energy and board-breaking system before impact. It can use mathematical tools to solve the impact. The energy conservation model is shown in Figure 1, where R1 is the radius, M1 is the mass, V0 is the breaking target board thickness of the rectangular board, and H is the rectangular board. During impact (t > 0), the impact force produces two deformations: (1) the contact deformation of the fragments and target board; (2) the horizontal displacement of the board, measured from the surface of the target board center. The center of the deformation in the board is near the center of the shock. The relationship between the impact power and contact deformation follows the Hertz law. The horizontal displacement of the target board is the sum of the bending deformation, Wb, of the board and horizontal shear deformation, WS. In addition, the membrane deformation is caused by stretching related to the target board displacement; if W/H < 0.2, the film effect can be ignored. Simultaneously, the target board was bent to investigate the signals generated by the sliding of the molecular crystal lattice; however, the signal of interference was too weak to test the kinetic energy of the chip. Therefore, its impact was also ignored. Hence, the displacement caused by shearing and deformation was only considered. Energy loss owing to material damping, surface friction, and high-end modular vibration was ignored. A standard Newton–Raphson algorithm was used to solve the collision equation. The maximum kinetic energy before the impact is for T = 0, and for T > 0, the target board-evasion system undergoes contact deformation, bending deformation, and film deformation, to store the deformation; then, the exhaustion of the exhausting target quality of the spray (stuffed or debris, etc.) and the remainder of the film occurs. A departure from the principle of total energy conservation, it is obtained from the energy conservation equation of the target board-evasion system as follows [21,22,23]:
(1)$$\frac{1}{2}M_{1}V_{0}{}^{2}=E_{C}+E_{bs}+E_{m}+\frac{1}{2}(M_{2}+M_{s})V_{2}^{2}$$
where $M_{1}$, $M_{2}$ and $M_{s}$ are the surplus quantity of the debris and mass of the fragments; $V_{2}$ is the speed of the film fragments and splash; and $E_{C}$, $E_{bs}$, $E_{m}$ are the storage energy corresponding to the contact, bending, and film deformations, respectively.

The corresponding force of the storage energy–deformation relationship is calculated as follows:(2)$$E_{c}=\frac{2}{5}\frac{\alpha^{3/5}}{n^{2/3}}$$
where α is the center of the target center and the center of the broken film close to each other, and n is the contact stiffness parameter, which depends on the material and geometric characteristics of the target board and the broken film.

The pressure of the target board can be divided into two components:(3)$$P=P_{bs}+P_{m}$$
where $P_{bs}$ is the bend and shear deformation dependent force and $P_{m}$ is the film-deformation-dependent force. Equation (3) can be simplified as follows:(4)$$P=K_{bs}\omega +K_{m}\omega^{3}$$
where $K_{bs}=(K_{b}K_{s})/(K_{b}+K_{s})$ is the effective stiffness of bending and shear, and $K_{b}$, $K_{s}$, $K_{m}$ are the bending strength, shear stiffness, and membrane constraints of the target, respectively.

The initial impact is unknown, and the contact radius between the impactor and the target plate $\alpha_{c}=h/2$; combining Equations (3) and (4), the following are obtained:(5)$$E_{bs}=\frac{1}{2}K_{bs}\omega^{2}$$
(6)$$E_{m}=\frac{1}{2}K_{m}\omega^{4}$$

Combined with the Equations (4)–(6), the following energy conservation formula is obtained:(7)$$M_{1}V_{0}^{2}=K_{bs}\omega^{2}+\frac{K_{m}\omega^{4}}{2}+\frac{4}{5}{\left[\frac{{\left(K_{bs}\omega +K_{m}\omega^{3}\right)}^{5}}{n^{2}}\right]}^{1/3}+(M_{2}+M_{s})V_{2}^{2}$$

If the remaining kinetic energy and spatter kinetic energy are known, the Newton–Raphson algorithm can be used to determine the break kinetic energy using w. Ignoring the friction of the target board, the turning inertia, and the shear effects, according to the assumption of the fragment hitting the target plate, the elastic collision model, based on the classical motion equation expressed by the transverse displacement, ω, per unit area of the elastic target plate incident on the axisymmetric time-varying load, Z(r,t), can be established as follows:(8)$$D{\left(\frac{\partial^{2}}{\partial r^{2}}+\frac{1}{r}\frac{\partial }{\partial r}\right)}^{2}\omega +2\rho h\frac{\partial^{2}\omega }{\partial t^{2}}=Z(r,t)$$
where $D=2h^{2}E/[3(1-\mu^{2})]$, ρ, 2h, E, μ are the density, thickness, and elastic constant of the target plate, respectively, and r and t are the radial coordinates and time, respectively. If the concentrated load F(t) acts on the origin of the wireless panel, Equation (8) can be transformed into

(9)$$\omega (r,t)=\frac{1}{8\pi \rho Bh}\int_{-\infty }^{t}\sin[\frac{r^{3}}{4B(t-u)}]\frac{du}{t-u}\int_{0}^{u}F(\eta )d\eta$$
where $B=\sqrt{D/2\rho h}$, and t is time. It can be observed that the qualitative point displacement and impact force become a certain functional relationship, and the target board method can be used as an amplitude parameter of the sound-transmission signal, that is, the impact force of the collision of the target board through the sound transmission signal and feature parameters can be characterized.

The signal recorded by the sound-transmitting sensor is the speed signal, so Equation (9) is used to obtain
(10)$$\frac{d\omega (r,t)}{dt}=\frac{1}{8\pi \rho Bh}\frac{dG(r,t-u)}{dt}\int_{0}^{u}F(\eta )d\eta$$

G(r,t−u) is $\int_{-\infty }^{t}\sin[r^{3}/4B(t-u)]/(t-u)du$. A functional relationship is observed between the acoustic transmission signal and the impact force.

It is assumed that the remaining speed after penetrating the target board is equivalent to splash fragments, which are virtually constant. According to reference [24], the remaining speed of the spherical fragment after penetrating the target board is expressed as follows:(11)$$V_{2}=V_{1}-\frac{1}{m}\int_{0}^{tf}Fdt$$

When and only when the condition $V_{2}-V_{1}/V_{1}<<1$, Equation (10) is established; the impact of the target board of the film fragments and perforation speed of the film fragments become a functional relationship.

Combined with Equations (7), (10) and (11), the initial kinetic energy and sound-transmission signal of the film fragments, as well as the remaining speed of the film fragment, exhibit a certain functional relationship. Therefore, to determine the broken sheet material and geometric shape, the smaller impact on the spatter and thermal energy loss during the perforation target board are ignored, and the initial kinetic energy of the broken film can be characterized using the sound-transmission signal.

## 3. Typical Evasion Hit Sound Emission Signal Numerical Simulation

### 3.1. Finite Element Model

To prove the functional relationship between the sound-transmission signal generated by the target board and the initial power of the film fragment, the ANSYS software was used for the numerical calculation of the process of typical broken sheets of aluminum alloy target boards. The speeds of the different Gaussian dots were obtained as a sound-transmission signal. Based on the symmetry of the spherical projectiles, a two-dimensional axisymmetric model was established. The typical fragment is simulated as spherical tungsten beads, and the target material is simulated as an AL7039 aluminum alloy plate. Figure 2 shows a two-dimensional axisymmetric model of a spherical fragment impacting a flat plate. The thickness and height of the target plate are 0.2 and 200 mm, respectively, and the fragment diameter is 8 mm. The numerical simulation algorithm adopted the Lagrange algorithm. The kinetic energy range of the fragment impacting the target plate is 375–6000 J, and its material parameters are listed in Figure 2. The acoustic emission acquisition point is set 75 mm away from the impact center of the target plate.

The simulation of the fragment high-speed impact experiment is based on the material model, which includes an equation of state, and a strength equation. The shock equation of state and the Johnson–Cook strength model describe fragments and target materials. The shock equation of state material impact velocity particle velocity relationship is expressed as follows:(12)$$U_{S}=C_{0}+Su_{p}$$
where *U_s_* is the impact speed, *u_p_* is the particle velocity, *S* is the slope of the relationship between *U_s_* and *u_p_*, and *C*_0_ is the body acoustic wave velocity.

The equivalent stress equation of the Johnson–Cook strength model is [21,22,23]:(13)$$\sigma_{y}=\left(A+B\varepsilon_{p}^{n}\right)\left[1+c\ln\varepsilon^{*}\right]\left(1-T^{*m}\right)$$
where *A*, *B*, *c*, and *n* are material constants,

$\varepsilon_{p}$ is the equivalent stress, is the $\varepsilon^{*}$ quorulent functional plastic strain rate, $T^{*m}$ and is the Quasi-quantity scheduled temperature.

The three terms on the right of the equation represent the effects of equivalent plastic strain, strain rate, and temperature, respectively, on flow stress.

The target sheet material parameters are listed in Table 1 [24], and the parameters of the film fragments are listed in Table 2 [25,26].

### 3.2. Sound Transmission Signal Time-Frequency Analysis

The above theoretical analysis shows that the acoustic emission signal can be used to estimate the initial kinetic energy of typical fragments when determining the material properties, as well as the geometry of fragments and target plates. The processing methods of acoustic emission signals include parameter and time-frequency analyses, whereas time-frequency analysis methods such as wavelet transform, HHT transform, and pseudo Wigner–Ville transform can be used to obtain the energy distribution of the acoustic emission signal in time and frequency simultaneously. This study analyzed the spectrum characteristics of the sound-transmission signal using the small wave bag analysis method. Because the time-domain characteristics of the acoustic emission signal generated by the fragment impacting the target plate rapidly increased and then fell gradually, it was impossible to use the symmetrical wavelet basis and db series wavelet basis to analyze the frequency band of the acoustic emission signal. The signal was decomposed and reconstructed according to the frequency band division of the wavelet decomposition. db8 was selected as the wavelet base, and the signal was decomposed to the fourth layer. The first 14 nodes of the optimal tree were analyzed. The acoustic emission signal in the frequency range of 0–2 MHz was divided into 14 frequency segments, and the interval range of each frequency segment was 62.5 kHz. The energy and energy proportion of each reconstructed signal were calculated as follows [12]:(14)$$E_{i,j}={\sum_{k=1}^{m}\left|x_{j,m}\right|}^{2}$$
where $E_{i,j}$ is the energy of each reconstruction signal and $x_{j,m}$ is the amplitude of the reconstructed signal discrete point.

(15)$$E_{toal}=\overset{2^{i}-1}{\underset{j=0}{\sum }}E_{i,j}$$(16)$$E_{j}=\frac{E_{i,j}}{E_{\mathrm{toal}\;}}$$where $E_{\mathrm{toal}}$ is the total energy of the signal and is $E_{j}$ the proportion of the total signal of each frequency band energy.

Combining Equations (14)–(16) can help one to obtain the energy changes of different frequency bands after signal wavelet decomposition. The time domain signal, wavelet time spectrum, and energy proportion of each frequency band within 30 µs of the impact acoustic emission signal 75 mm from the impact center of the target plate were obtained under the action of the selected kinetic energies of 960, 1815, 3375, and 4860 J.

Figure 3b, Figure 4b, Figure 5b and Figure 6b show that the acoustic emission energy is predominantly concentrated in the low-frequency band. With increasing initial kinetic energy, the acoustic emission signal frequency increases, and the high frequency does not exceed 200 kHz.

Figure 3c, Figure 4c, Figure 5c and Figure 6c show that the signal energy proportion in the frequency bands of 0–62.5, 62.5–125, 125–187.5, and 187.5–250 kHz is the highest, and the energy proportion is concentrated in the 250 kHz frequency band. With increasing initial kinetic energy, the acoustic emission signal frequency increases to a high frequency. As mentioned above, the difference of acoustic emission signals corresponding to different impact kinetic energy is predominantly concentrated in the energy difference of the high-frequency band within 250 kHz, that is, the signal analysis of the first three frequency bands after wavelet decomposition.

To quantitatively analyze the differences in acoustic emission signals of target penetration under fragment velocity damage, the total energy and typical fragment kinetic energy of acoustic emission signals in the frequency ranges of 62.5–125, 125–187.5, and 187.5–250 kHz decomposed using wavelet were statistically analyzed. The mathematical relationship of typical evasion initial kinetic energy and sound emission wavelet total energy was determined. Figure 7 shows the wavelet energy of the evasion hit signal with speed change, based on the statistical method to fit the evaluation and wavelet energy shown in Equation (17).
(17)$$y=6.58x^{-0.28}-1.077$$
where x is evapogenic energy and y is the total energy value of the frequency section of the wavelet decomposition. The sum of the squares of errors is 0.5484, and the determination coefficient is 0.9891. It can be observed that within a certain error range, the wavelet energy in the 250 kHz frequency band has an exponent relationship with the fragment kinetic energy, and the 8 mm typical fragment kinetic energy can be characterized by the wavelet energy in the 250 kHz frequency band.

## 4. Typical Evasion Hit Target Board Test Verification

### 4.1. Evasion Test Target

The acoustic emission test of fragments impacting the target plate is performed in an explosion field, and the experimental conditions are the same as those in the simulation. The fragment driving device emits high-speed spherical tungsten beads, which are tested using a light curtain target velocity measuring device and an acoustic emission acquisition device. As shown in Figure 8a, the light curtain target speed is placed in front of the aluminum alloy target, and the light-curtain target spacing, s, is 1 m. The time, Δs, of fragments passing through adjacent targets was recorded; the formula, s/Δt, was used to obtain the average velocity between adjacent light curtain targets. The instantaneous velocity when hitting the target plate was obtained as the average velocity of multiple groups of light curtain targets and the fragment velocity attenuation formula. The fragment kinetic energy calculated from the impact velocity obtained by the light curtain target was verified and compared with the fragment kinetic energy measured using the acoustic emission technology. When the fragment passed through the light curtain target and struck the aluminum alloy target plate, the pvdf piezoelectric film pasted on the back of the target plate received the acoustic emission signal; the two pvdf sensor piezoelectric constant was 43.94 PC/N·cm^2^; and the area was 6 mm × 6 mm and 6 mm × 12 mm, respectively. Simultaneously, the charge amplifier and data acquisition system were connected to form a test system. The composition of the fragment impact target acoustic emission test system is shown in Figure 9. The entire test covers the impact kinetic energy of 375–6000 J, and the impact is performed randomly in this range. A total of three tests were conducted, and the test conditions are shown in Figure 8b.

### 4.2. Comparative Analysis and Numerical Calculation Results

The periphery of the thin target is fixed, and no gradient between stress and strain is present in the thickness direction. To a certain extent, the attenuation of acoustic emission signal during the propagation of the target medium can be ignored. The acoustic emission signal obtained by the typical spherical fragment impacting the aluminum alloy thin target plate six times is subjected to 30 kHz high-pass filtering. The time-frequency analysis after the first three filters is shown in Figure 10 below.

Figure 10 shows the three-dimensional time-frequency diagram of the acoustic emission signal collected by the fragment impacting the thin target. It is evident from the time-frequency diagram that the signal energy is predominantly concentrated in 250 kHz in 20 us. Moreover, it is evident from the time-frequency analysis diagram of the first and second engine tests that there is no high-frequency component, while the time-frequency diagram of the third engine test shows that its frequency component moves to a high-frequency band. According to the conclusion drawn above, the increase in fragment kinetic energy causes an increase in its frequency, that is, the fragment kinetic energy of the third engine is higher than those of the first and second engines. The experimental results obtained by extracting the wavelet energy in the 250 kHz frequency band through wavelet packet analysis and the model established based on the simulation data are listed in Table 3.

The timing signal acquisition of the light curtain target velocity measuring device was determined at the starting time at the muzzle. The fragments obtained by the data acquisition system passed through the target signal of the light curtain target. The test results are listed in Table 4.

A comparison of the speed measurement of the light curtain target speed measuring device and the speed measurement based on acoustic emission signal is shown in Table 5.

It can be seen that the acoustic emission signal generated by fragments penetrating the thin target contains information related to kinetic energy. This is a new method to calculate the fragment kinetic energy and analyze the damage mechanism of fragments penetrating the target, from the perspective of acoustic emission signal. With the increasing initial kinetic energy of the film fragments, the main frequency component of the signal moves from a low-frequency zone to a high-frequency zone, resulting in a more complicated frequency component. The low-frequency band small waves of the sound transmission signal decrease with increasing speeds, and the proportion of high-frequency waves to small waves increases with the increasing rate. Within the allowable error range, the kinetic energy obtained using the light curtain target test according to the classic kinetic energy formula can be represented using a sound-transmission feature parameter wave-energy model.

## 5. Conclusions

In this study, the interaction of stress waves during the penetration of thin targets by fragments was theoretically analyzed from the perspective of the dynamic response of the target fracture. The impact acoustic emission signals of the target under different kinetic energies were obtained through numerical simulation, and the signal characteristics were analyzed using wavelet packet technology and a statistical method. Combined with the light curtain target comparison test, the following conclusions were drawn:

(1) During the high-speed fragment penetration of a thin target, theoretically, a quantitative relationship exists between the amplitude of the acoustic emission signal and the initial kinetic energy of the fragments under the impact conditions when determining the material properties and geometry of the fragments and target plates.

(2) The relationship between the AE signal and the fragment kinetic energy was simulated and analyzed. With the increasing initial kinetic energy of fragments, the main frequency component of the signal will move from a low- to a high-frequency band, resulting in a more complex frequency component. The low-frequency band wavelet energy of the acoustic emission signal decreases with increasing velocity, and the high-frequency band wavelet energy is positively correlated with the fragment velocity. Based on statistical analysis, it was observed that the wavelet energy of acoustic emission signals generated by fragments with the same diameter penetrating the thin target at different initial velocities is an idempotent function with fragment kinetic energy.

(3) The test results of the fragment shooting range demonstrated that the mathematical model of the acoustic emission signal wavelet energy and the projectile kinetic energy exhibited high fitting accuracy. Three acoustic emission kinetic energy tests were analyzed using the light curtain target velocity measurement comparison test. The measurement errors were 3%, 3.7%, and 3%, respectively.

## Figures and Tables

**Figure 1 sensors-22-05914-f001:**
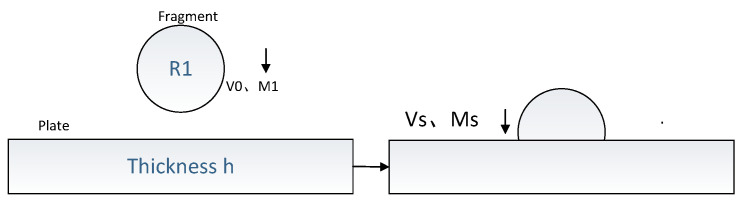
Fragment impacting target plate.

**Figure 2 sensors-22-05914-f002:**
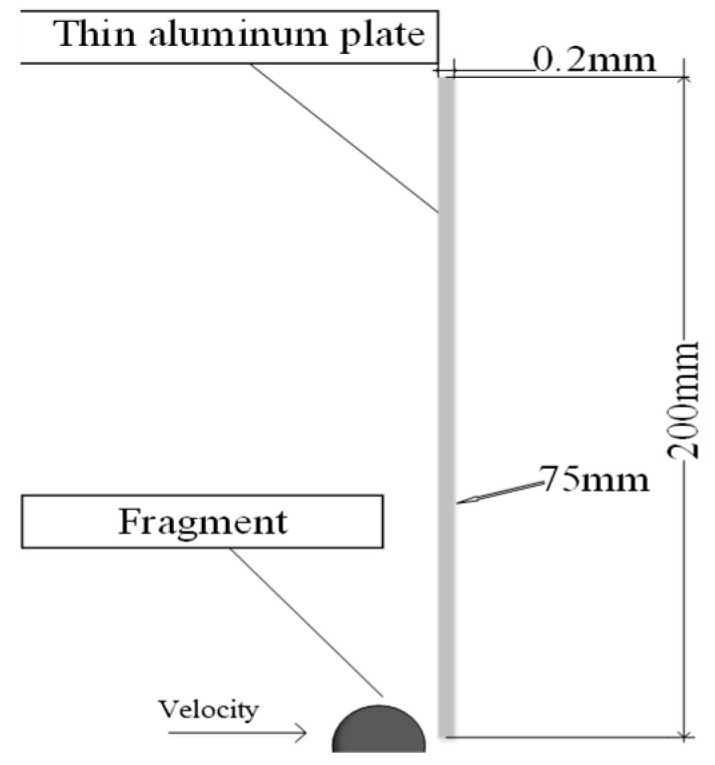
Numerical simulation model of broken film impact target.

**Figure 3 sensors-22-05914-f003:**
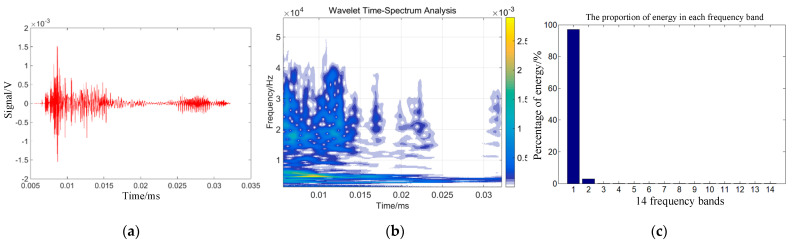
Time-frequency analysis of target sound emission signals under 960 J. (**a**) Acoustic emi-sion time domain waveform. (**b**) Sound emission wavelet time spectrum. (**c**) Acoustic transmitted signals for each frequency band.

**Figure 4 sensors-22-05914-f004:**
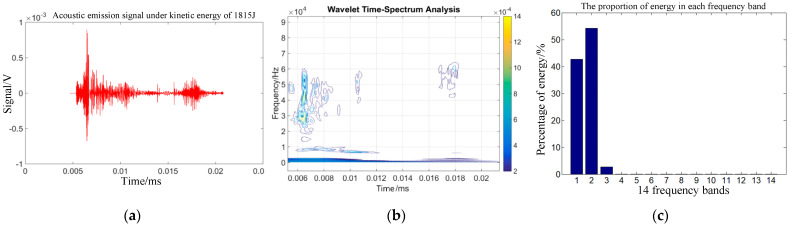
Time-frequency analysis of target sound emission signals under 1815 J kinetic energy. (**a**) Acoustic emission time domain waveform. (**b**) Sound emission wavelet time spectrum (**c**) Acoustic transmitted signals for each frequency band.

**Figure 5 sensors-22-05914-f005:**
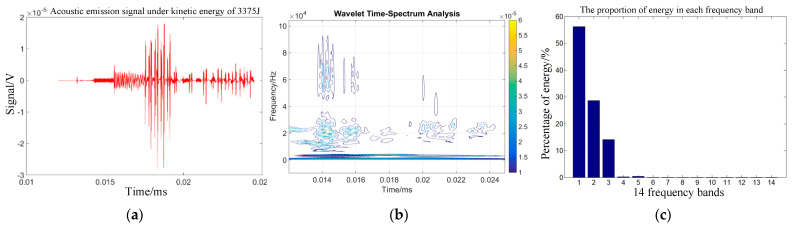
Time-frequency analysis of target sound emission signals under 3375 J kinetic energy. (**a**) Acoustic emission time domain waveform. (**b**) Sound emission wavelet time spectrum. (**c**) Acoustic transmitted signals for each frequency band.

**Figure 6 sensors-22-05914-f006:**
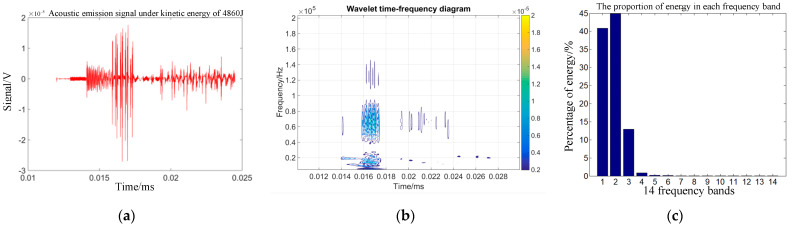
Time-frequency analysis of target sound emission signals under 4860 J. (**a**) Acoustic emi-sion time domain waveform. (**b**) Sound emission wavelet time spectrum. (**c**) Acoustic transmitted signals for each frequency band.

**Figure 7 sensors-22-05914-f007:**
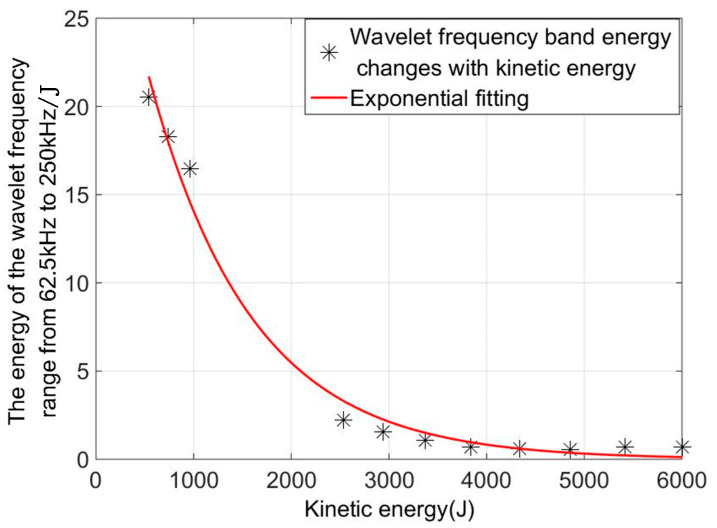
Wavelet frequency band energy value with kinetic energy change.

**Figure 8 sensors-22-05914-f008:**
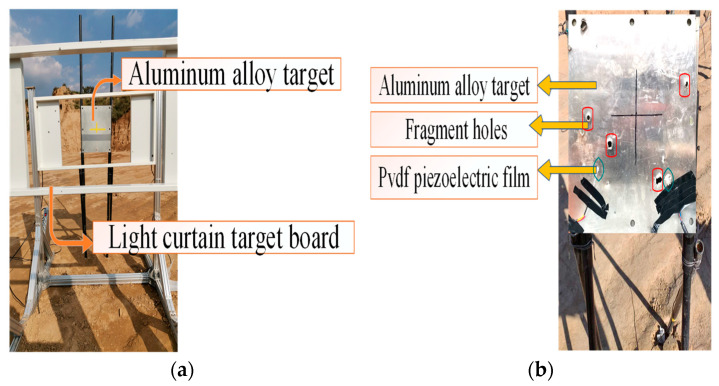
Evasion of the slurry target. (**a**) Evasion hit target board experimental device. (**b**) Evasion of the target results.

**Figure 9 sensors-22-05914-f009:**
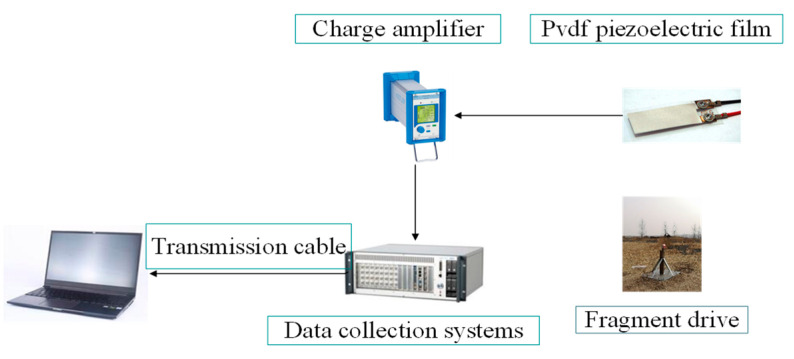
Schematic of interval impact target sound emission experiment system.

**Figure 10 sensors-22-05914-f010:**
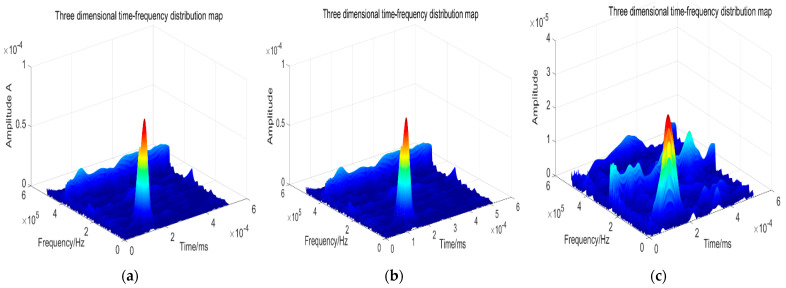
Target sound emission signal three-dimensional time frequency analysis. (**a**) First three-dimensional time-frequency map. (**b**) Second three-dimensional time-frequency map. (**c**) Third three-dimensional time-frequency map.

**Table 1 sensors-22-05914-t001:** Target material parameters.

Material	ρ/(g/cm^3^)		E/GPa	*u*
Aluminum 7039	2.77		79	0.33
A/MPa	B/MPa	C	n	m
337	343	0.01	0.41	1

**Table 2 sensors-22-05914-t002:** Evasion material parameters.

Material	ρ/(g/cm^3^)	E/GPa	*u*
Tungsten	1.76	350	0.284

**Table 3 sensors-22-05914-t003:** Disturbed experiment target sound signal calculation results.

Test	Target Sheet Material	Fragment Material	Wavelet Energy	Fragment Kinetic Energy (J)	Target Status
Test 1	Aluminum 7039	Tungsten Alloy	3.114	2598	Perforation
Test 2	Aluminum 7039	Tungsten Alloy	3.117	2597	Perforation
Test 3	Aluminum 7039	Tungsten Alloy	1.917	3112.2	Perforation
Test 4	Aluminum 7039	Tungsten Alloy	1.893	3143.5	Perforation
Test 5	Aluminum 7039	Tungsten Alloy	1.693	3289.5	Perforation
Test 6	Aluminum 7039	Tungsten Alloy	1.034	3778.4	Perforation

**Table 4 sensors-22-05914-t004:** Light curtain target speed device experiment.

Test	Light Curtain Target Spacing (m)	Target Plate Interval (ms)	Velocity at Target (m/s)	Fragment Quality (g)	Fragment Kinetic Energy (J)
Test 1	1	0.748	1336.3	3	2678.6
Test 2	1	0.774	1292.4	3	2505.4
Test 3	1	0.685	1460.6	3	3200
Test 4	1	0.665	1503.8	3	3240.3
Test 5	1	0.486	2057.6	3	3308.1
Test 6	1	0.412	2427.18	3	3879

**Table 5 sensors-22-05914-t005:** Light curtain target speed and target results compared with target.

	Kinetic Energy/J		
Test	Light Curtain Target Experiment	Wavelet Energy Calculation	Error/%
Test 1	2598	2678.6	3
Test 2	2597	2505.4	3.7
Test 3	3112.2	3200	2.7
Test 4	3143.5	3240.3	2.99
Test 5	3289.5	3308.1	0.5
Test 6	3778.4	3879	2.59

## Data Availability

The raw/processed data required to reproduce these findings cannot be shared at this time as the data also forms part of an ongoing study.

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
