# Peer review of "Typical Fragment Kinetic Energy Assessment Based on Acoustic Emission Technology"

_sensors, 2022, doi:10.3390/s22155914_

Round 1

Reviewer 1 Report

1. Fig 3~6 need to be improved. It is not clear.  X axis coordinate title of Fig 3~6 (c) is inconsistent.

2. Fig 3~6 (b) 3 does not need to show time-frequency analysis greater than 300kHz.

3. How large is the fragment volume selected in this paper, and does the fragment volume affect the detected acoustic emission signal?

4.What is the fixing method of aluminum alloy target?

5.Do other debris holes on the aluminum alloy target affect the detected acoustic emission signal?

Author Response

Dear reviewer, thank you very much for your support and hard work on my paper, and thank you very much for your valuable review comments. I will carefully revise the paper according to your review comments and explain it to you point by point. As for the language of the paper you mentioned, a certain degree of revision is required. At present, we have carefully checked and revised the language description of the full text, and invited a professional inspection agency to check the language of the paper, avoiding as much as possible language errors. The language editing certificate is attached at the end of the author's reply to the reviewer's comments. On behalf of all the authors, I would like to extend my heartfelt thanks to you again, I wish you a happy life and smooth work!

  • Comment 1:Fig 3~6 need to be improved. It is not clear.  X axis coordinate title of Fig 3~6 (c) is inconsistent.
  • Response:Dear reviewer, as you said, Figures 3 to 6 in the paper are not clear enough, and there are inconsistencies in the labelling of the coordinate axes. I am very sorry for the question you mentioned. At present, Figures 3 to 6 have been redrawn. The drawn figures are as follows, and have been modified in the paper.

  • Comment 2:Fig 3~6 (b) 3 does not need to show time-frequency analysis greater than 300kHz.
  • Response:Dear reviewer, thank you very much for bringing up this issue. At present, according to your review comments, the content above 300kHz in Fig 3~6 (b) in the paper has been removed, and the revised content was added in the paper.

  • Comment 3:How large is the fragment volume selected in this paper, and does the fragment volume affect the detected acoustic emission signal?
  • Response:Dear reviewer, the fragment is a spherical fragment with a diameter of 8mm, and its volume is 67.020643mm^3. This paper only analyzes the relationship between fragment kinetic energy and acoustic emission signal under typical fragments. The mathematical model between fragment kinetic energy and acoustic emission signal mainly studies the model establishment and test verification of 8mm typical fragment kinetic energy, the influence of projectile volume and shape on the mathematical model, and establishes mathematical models for the acoustic emission signal and fragment kinetic energy of projectiles with different shapes. The power-exponential function relationship under the typical fragment mathematical model can be further revised.

  • Comment 4:What is the fixing method of aluminum alloy target?
  • Response:Dear reviewer, the question you mentioned is indeed very critical for the accurate measurement of acoustic emission signals. During the test, we made holes at the four corners of the aluminum alloy target, and installed the aluminum alloy target on the bracket by bolting, so that the aluminum alloy target and the mounting bracket became a whole.

  • Comment 5:Do other debris holes on the aluminum alloy target affect the detected acoustic emission signal?
  • Response:Dear reviewer, the remaining holes on the aluminum alloy target will affect the acoustic emission signal when the fragment hits the target to a certain extent, but the impact is very small. Therefore, from the general rule, the influence of the remaining holes on the aluminum alloy target on the acoustic emission measurement results can be ignored.

         Once again, on behalf of all authors, I would like to express my heartfelt thanks to the referee for your hard work and valuable comments on my paper. It is because of your valuable review comments that the overall quality of the paper has been improved. I wish you a happy life and smooth work!

Language Editing Certificate

Reviewer 2 Report

In the opinion of the referee the English of the Paper is unclear and needs to be improved. Especially the noun transmit should be replaced with  ' transfer'. The remarks on the Battle of the Battle are unnecessary inside the scientific report.

The quality of the figures 3, 4,5 and 6 should be improved by the increasing of the resolution of the applied scans of the images.

In line 90 there is a typo 'impaact' instead of 'impact'.

When comparinh Tables 1 and 2 some parameters are listed as symbols there and some are described using their full names.

In line 281 it should be stated that the piezoelectric CONSTANT not 'media' is 43.94 pC /N x m2 . In the line 261 the term 'idempotent' should be replaced with 'exponent'  [relationship].

Author Response

Dear reviewer, thank you very much for your support and hard work on my paper, and thank you very much for your valuable review comments. I will carefully revise the paper according to your review comments and explain it to you point by point. As for the language of the paper you mentioned, a certain degree of revision is required. At present, we have carefully checked and revised the language description of the full text, and invited a professional inspection agency to check the language of the paper, avoiding as much as possible language errors. The language editing certificate is attached at the end of the author's reply to the reviewer's comments. On behalf of all the authors, I would like to extend my heartfelt thanks to you again, I wish you a happy life and smooth work!

  • Comment 1:In the opinion of the referee the English of the Paper is unclear and needs to be improved. Especially the noun transmit should be replaced with  ' transfer'. The remarks on the Battle of the Battle are unnecessary inside the scientific report.
  • Response:Dear reviewers, I am very sorry that the expert review is affected by the inappropriate use of language and grammar in the writing process of the paper. Based on your comments, the full text has been carefully revised, and native English speakers have been invited to review the paper to ensure that there are as few language errors as possible. Thank you very much for this question. I will improve my language skills in the subsequent paper writing process.

  • Comment 2:The quality of the figures 3, 4,5 and 6 should be improved by the increasing of the resolution of the applied scans of the images.
  • Response:Dear reviewer, as you said, Figures 3 to 6 in the paper are not clear enough, and there are inconsistencies in the labelling of the coordinate axes. I am very sorry for the question you mentioned. At present, Figures 3 to 6 have been redrawn. The drawn figures are as follows, and have been modified in the paper.

  • Comment 3:In line 90 there is a typo 'impaact' instead of 'impact'.
  • Response:Dear reviewer, I am very sorry for the mistakes made by the author due to the author's inattentiveness in the writing process. At present, the misspelled words in the paper have been changed according to your comments.

  • Comment 4:When comparinh Tables 1 and 2 some parameters are listed as symbols there and some are described using their full names.
  • Response:Dear reviewer, this question you mentioned is indeed very pertinent. Indeed, as you said, the same parameter names should be the same, and the description of the inconsistent parameters in Table 1 and Table 2 in the paper has been revised.

  • Comment 5:In line 281 it should be stated that the piezoelectric CONSTANT not 'media' is 43.94 pC /N x m2 . In the line 261 the term 'idempotent' should be replaced with 'exponent'  [relationship].
  • Response:Dear reviewer, thank you very much for mentioning this issue, which I failed to notice during the writing process. The paper has been revised according to your review comments.

        Once again, on behalf of all authors, I would like to express my heartfelt thanks to the referee for your hard work and valuable comments on my paper. It is because of your valuable review comments that the overall quality of the paper has been improved. I wish you a happy life and smooth work!

Language Editing Certificate

Reviewer 3 Report

Review of Fei Shang et al., Typical fragment kinetic energy assessment based on acoustic emission technology

 This manuscript describes an acoustic emission technology-based method to evaluate the fragment kinetic energy. Numerical simulation and wavelet energy of acoustic emission signal were investigated by wavelet packet theory and mathematical model. Acoustic emission fragment kinetic energy suggests a new method for directly measuring fragment kinetic energy. New exciting results are presented about the relation between acoustic wave energy and kinetic energy of fragments, but some minor points have to be clarified to improve the paper's scientific content. I would recommend a minor revision.

Specific comments

1.      Although the frequencies of acoustic emission signals are similar, sound waves have several types, such as acoustic waves, whistlers, etc. Please discuss the possible waves for future work.

2.      When authors want to use equations and numerical models, references should be cited. For example, references to the Johnson-Cook strength model are needed in line 193 on page 5. Please add references in an equation, model, software, and so on.  

3.      Typos: “impaact” in line 82 on page 2; 30 us in line 228 on page 7.

4.      Rearrange the frequency scale in Figures 3(b) and 4(b).

5.      Please add the unit on y-axis in Figure 7.

Author Response

Dear reviewer, thank you very much for your support and hard work on my paper, and thank you very much for your valuable review comments. I will carefully revise the paper according to your review comments and explain it to you point by point. As for the language of the paper you mentioned, a certain degree of revision is required. At present, we have carefully checked and revised the language description of the full text, and invited a professional inspection agency to check the language of the paper, avoiding as much as possible language errors. The language editing certificate is attached at the end of the author's reply to the reviewer's comments. On behalf of all the authors, I would like to extend my heartfelt thanks to you again, I wish you a happy life and smooth work!

  • Comment 1:Although the frequencies of acoustic emission signals are similar, sound waves have several types, such as acoustic waves, whistlers, etc. Please discuss the possible waves for future work.
  • Response:Dear reviewer, thank you very much for bringing up this issue. According to your review comments and the current research progress of the research group, in the follow-up research, we will introduce the information of impact speed, displacement and acceleration as the characterization parameters of the acoustic emission signal, so as to further analyze the impact of the fragments on the target plate. Acoustic emission signal, improve the measurement accuracy of acoustic emission signal and the accuracy of the established model.

  • Comment 2:When authors want to use equations and numerical models, references should be cited. For example, references to the Johnson-Cook strength model are needed in line 193 on page 5. Please add references in an equation, model, software, and so on. 
  • Response:Dear reviewer, thank you very much for your mention of this problem. This is a problem caused by the author's lack of rigorous consideration in the process of writing the paper. At present, the relevant literature and materials have been reviewed, the paper has been checked, and citation marks have been added in the corresponding positions. In subsequent essay writing, I will improve my writing skills to avoid such mistakes.

  • Comment 3:Typos: “impaact” in line 82 on page 2; 30 us in line 228 on page 7.
  • Response:Dear reviewer, I am very sorry for the mistakes made by the author due to the author's inattentiveness in the writing process. At present, the misspelled words in the paper have been changed according to your comments.

  • Comment 4:Rearrange the frequency scale in Figures 3(b) and 4(b).
  • Response:Dear reviewer, thank you very much for your comments on my paper. As you said, the frequency labels in the original Figures 3(b) and 4(b) are inappropriate, and the frequencies in Figures 3(b) and 4(b) have been relabeled. Graphics have been added to the paper.

  • Comment 5:Please add the unit on y-axis in Figure 7.
  • Response:Dear reviewer, I am very sorry that due to the author's negligence in the writing process, the ordinate in Figure 7 has not added units. According to your review comments, Figure 7 has been revised and the revised figure has been added to the paper. middle.

         Once again, on behalf of all authors, I would like to express my heartfelt thanks to the referee for your hard work and valuable comments on my paper. It is because of your valuable review comments that the overall quality of the paper has been improved. I wish you a happy life and smooth work!

Language Editing Certificate
